# Heterogenous electromediated depolymerization of highly crystalline polyoxymethylene

Yuting Zhou [1,2], Joaquín Rodríguez-López [1,2,3] & Jeffrey S. Moore [1,2,3]

Post-consumer plastic waste in the environment has driven the scientific community to develop deconstruction methods that yield valued substances from these synthetic macromolecules. Electrocatalysis is a well-established method for achieving challenging transformations in small molecule synthesis. Here we present the first electro-chemical depolymerization of poly-oxymethylene—a highly crystalline engineering thermoplastic (Delrin®)—into its repolymerizable monomer, formaldehyde/1,3,5-trioxane, under ambient conditions. We investigate this electrochemical deconstruction by employing solvent screening, cyclic voltammetry, divided cell studies, electrolysis with redox mediators, small molecule model studies, and control experiments. Our findings determine that the reaction proceeds via a heterogeneous electro-mediated acid depolymerization mechanism. The bifunctional role of the co-solvent 1,1,1,3,3,3-hexafluoro-2-propanol (HFIP) is also revealed. This study demonstrates the potential of electromediated depolymerization serving as an important role in sustainable chemistry by merging the concepts of renewable energy and circular plastic economy.

Plastics, as one of the most demanded synthetic materials, have reached a global annual production of 367 metric tons in 2020[1]. Strikingly, nearly 80% of the annually produced plastics are turned into waste, equivalent to ~290 metric tons every year[2]. In contrast to biomass with its short carbon cycle, carbon in commodity polymers primarily depend on petroleum, a non-renewable fossil fuel. Due to their high molecular weight and relatively inert linkages (C−C, C−O bonds), the life span of these synthetic macromolecules is generally persistent and non-circular[3]. Currently, the end-of-life management of synthetic plastics typically consists of mechanical recycling (a form of down-cycling), which cleaves synthetic macromolecules through high force. In doing so, the thermomechanical properties of the resulting materials are diminished, leading to limited applications of the recycled material[4].

In comparison to mechanical recycling, chemical recycling to monomer (CRM) is widely regarded as a more promising recycling strategy for these post-consumer plastics[5,6]. An example is the chemolysis of poly(ethylene terephthalate) (PET), but CRM of PET generally requires energy-intensive high temperature (120–180 °C) processes[7-9]. Recently, in addition to chemical recycling, chemical upcycling has gained numerous attention in the post-synthetic functionalization of commodity polymers[10-13]. However, most reported upcycling methods often proceed in extreme reaction conditions (120–500 °C, >10 bar), especially for high-performance plastics that exhibit high crystallinity (polyethylene (PE)[14-18], polypropylene (PP)[19], PET[20-22], polycarbonate (BPA-PC)[23,24]). To date, mild depolymerization strategies of commodity polymers have only been found to be effective for polystyrene (PS), an amorphous material (Fig. 1a)[25-27].

Post-synthetic functionalization/depolymerization of commonly known synthetic plastics (PE, PET, and PS) has reached significant breakthroughs over the past 5 years. However, very limited studies have been reported on the chemical recycling/upcycling of another

[1]Beckman Institute for Advanced Science and Technology, University of Illinois at Urbana–Champaign, Urbana, IL, USA. [2]Joint Center for Energy Storage Research, Argonne National Laboratory, 9700 South Cass Avenue, Lemont, IL 60439, USA. [3]Department of Chemistry, University of Illinois at Urbana–Champaign, Urbana, IL 61801, USA. ✉e-mail: jsmoore@illinois.edu

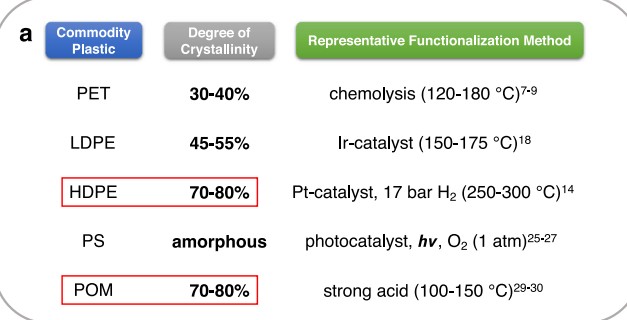

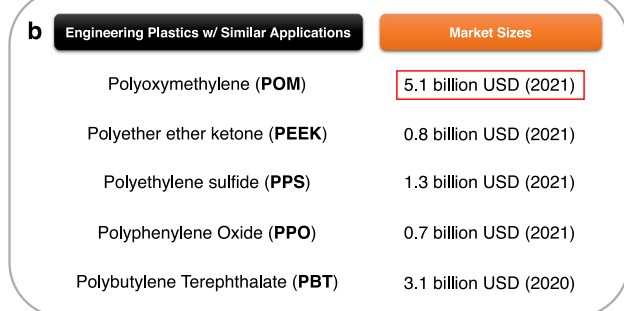

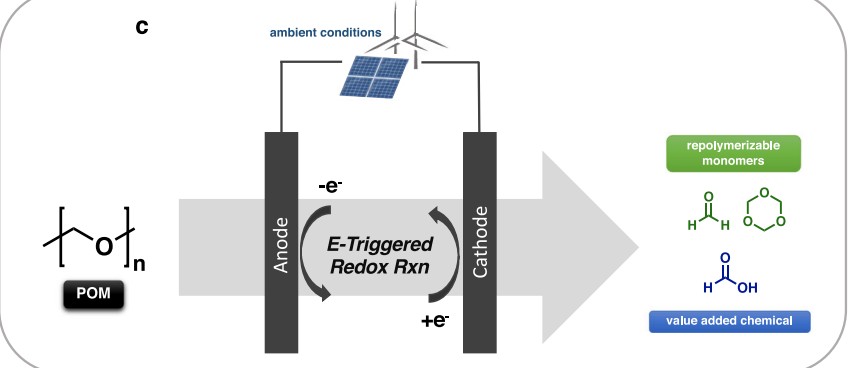

**Fig. 1 | Reported functionalization strategies of different commodity polymers and schematic illustration of E-mediated depolymerization. a** Degree of crystallinity[42] and typical functionalization approaches of different commodity polymers. **b** Market sizes of five high-performance engineering plastics with shared applications[28,43–46]. **c** E-mediated redox reaction couples: renewable energy & circular plastic economy.

high-performance polymer whose use is growing in popularity. Poly-oxymethylene (POM), also known as Delrin®, has been identified as one of the most crystalline engineering synthetic plastics (Fig. 1a). The reported market size of POM in 2021 reached 5.1 billion USD, with an estimated annual growth rate of 5.5% (from 2022 to 2028)[28]. POM is widely used as precise parts (ex., mechanical gear) in the automobile industry, owing to its superior mechanical properties (e.g., minimal friction, high dimensional stability, and excellent stiffness). Meanwhile, it held a significant market share among high-performance engineering plastics with similar applications (Fig. 1b). Similar to high-density polyethylene (HDPE), the high degree of crystallinity of POM introduces substantial challenges to its depolymerization. Currently, the reported chemical recycling/upcycling strategies for POM are limited to depolymerization via strong inorganic acids or expensive Lewis acids at elevated temperatures (100–150 °C) in a batch system[29–31]. Recently, electrocatalysis, as a versatile chemical strategy, has enabled remarkable innovations in synthetic methodology development[32], electro-editing of amorphous materials[33,34], and programmable deconstruction of redoxmers in flow battery system[35,36], but has yet to find applications in the deconstruction of highly crystalline materials. We conjectured that POM deconstruction is susceptible to an electromediated depolymerization.

Herein, a heterogeneous electromediated depolymerization of polyoxymethylene is reported. In the present investigation, POM is fully depolymerized into its monomeric forms (formaldehyde and 1,3,5-trioxane) under ambient conditions via potential control at an electrode (Fig. 1c). Compared to conventional chemical recycling/upcycling methods, electromediated depolymerization is mild (room temperature), energy-efficient, and robust. It does not require air-free techniques or dry solvents. In addition, free protons are generated via anodic oxidation and act as intermediates that catalyze the chain depolymerization, thus, circumventing the handling of strong inorganic acids. More importantly, electricity, the central reagent, is quickly becoming a renewable energy source via solar, wind, and

hydroelectric generation. We, therefore, demonstrate a sustainable energy-driven circular plastic economy.

## Results

### The crucial role of solvent in electro-depolymerization of POM

Like most high-performance engineering thermoplastics, poly-oxymethylene (POM) experiences severe solubility limits. Its highly regularly packed acetal chains offer unique mechanical properties[37]. Also, due to this ordered molecular structure, post-polymerization modification of POM is particularly challenging. POM is insoluble in most common organic solvents, especially ones that are commonly used in electrochemistry (see Supplementary Fig. 2). However, POM fully dissolves in 1,1,1,3,3,3-hexafluoro-2-propanol (HFIP) at room temperature. Compared to common polar protic solvents, the hexafluoro-substituted 2-propanol exhibits unique physical and chemical properties. In 2006, Berkessel and co-workers reported the strong amplification of the H-bonding ability of HFIP via aggregation[38]. The strong H-bonding donating ability of HFIP complements the poly-acceptor acetal chain. We suggest that the hydrophobic $CF_3$ moieties and the clustering behavior of HFIP further prevent the collapse of the polymer chain, enabling the complete dissolution of POM (Fig. 2a). To minimize the amount of the hexa-fluorinated solvent, HFIP is introduced as a co-solvent in the electro-depolymerization of polyoxymethylene.

Protic solvent isopropyl alcohol (more amphiphilic than water) was first chosen as the major co-solvent to carry out the electrolysis of polyoxymethylene. We expected isopropanol to function as a proton donor (by anodic oxidation). In an undivided cell setting, through cathodic reduction, hydrogenation of formaldehyde was initially expected to drive the depolymerization. However, as shown in Table 1 (Entry 1), no depolymerization of POM was observed using the IPA/HFIP solvent combination, despite continuous electrolysis for extended run times. The experiment conducted at higher applied potential (cell potential: 10 V) using IPA-HFIP as solvent combo also

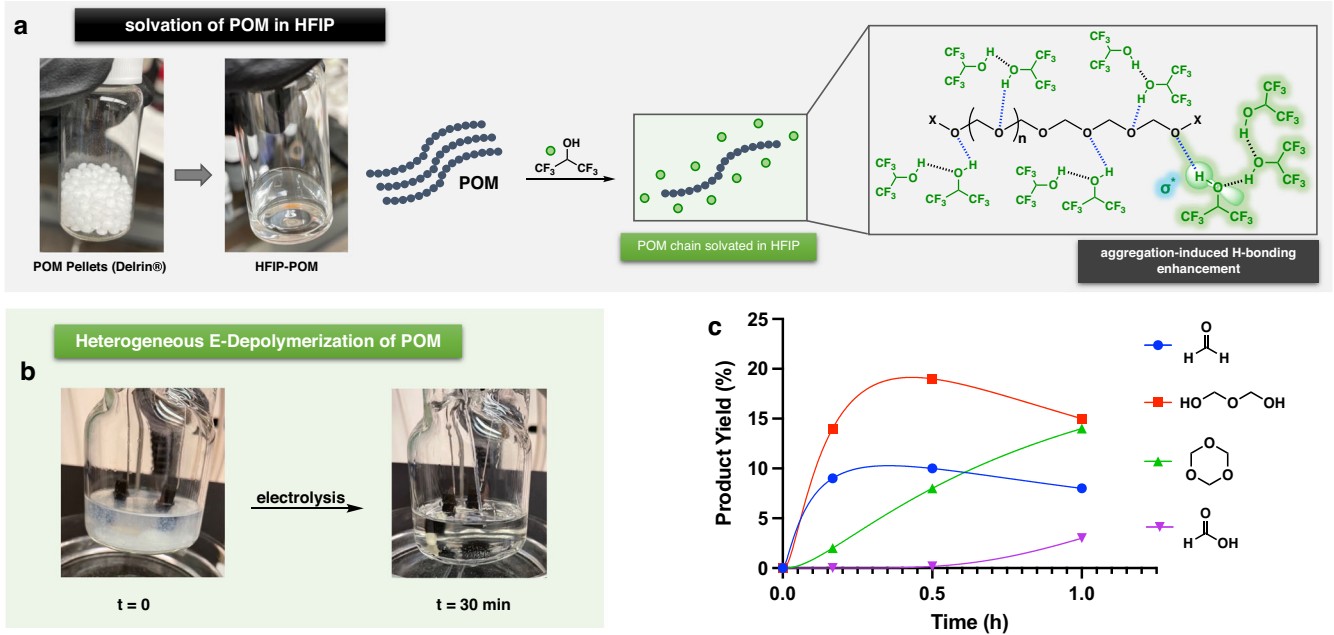

**Fig. 2 | The central role of HFIP: a solvent component that solvates POM.**
**a** 1,1,1,3,3,3-Hexafluoro-2-propanol (HFIP): an effective solvent for poly-oxymethylene (POM), the clustering of HFIP enabled the decrease of $\sigma^{*}_{OH}$ energy[38]. **b** Images of an undivided cell before (left) and after (right) the electrolysis. **c** Reaction profile of POM bulk electrolysis at 3.5 V (60 °C), 0.1 M LiClO$_4$ in CH$_3$CN:HFIP (26:4).

## Table 1 | Solvent impact

| Entry | Solvent | Applied potential vs. Ag/AgCl (V) | Temperature | Time | Product yield | | | |
|---|---|---|---|---|---|---|---|---|
| | | | | | I | II | III | IV |
| 1 | IPA:HFIP (26:4) | 3.5 | 60 °C | 2 h | No depolymerization | | | |
| 2 | H$_2$O:HFIP (26:4) | 3.5 | 60 °C | 2 h | No depolymerization | | | |
| 3 | DMF:HFIP (26:4) | 3.5 | 60 °C | 2 h | No depolymerization | | | |
| 4 | CH$_3$CN:HFIP (26:4) | 3.5 | 60 °C | 30 min | 10% | 19% | 8% | Trace |
| 5 | CH$_3$CN | 3.5 | 60 °C | 2 h | No depolymerization | | | |
| 6[a] | CH$_3$CN:HFIP (26:4) | 0 | 60 °C | 2 h | No depolymerization | | | |
| 7 | CH$_3$CN:HFIP (26:4) | 3.5 | rt | 2 h | 5% | 30% | 40% | 4% |
| 8[b] | CH$_3$CN | 3.5 | rt | 2 h | No depolymerization | | | |

Product yields were determined by ¹H NMR spectroscopy using mesitylene as standard reference. For details see supplementary information page S7.
[a]Here, 0 V referred to cell potential and no depolymerization with an extended run time of 24 h.
[b]No depolymerization with an extended run time of 8 h.

did not lead to any POM depolymerization (see Supplementary Fig. 11). From the NMR evidence (see Supplementary Figs. 10–12), continuous oxidation of isopropyl alcohol occurred, as shown by the accumulation of acetone. IPA does indeed function as a proton donor; however, it did not assist in the depolymerization of POM. When DI water was used as the major electrolysis solvent, the same phenomenon was observed and no depolymerization occurred (Table 1, Entry 2).

The polar aprotic solvent DMF (Table 1, Entry 3) was also found to be ineffective; upon electrolysis, DMF oxidation occurred. More redox-stable acetonitrile was then used with HFIP. As shown in Table 1 (Entry 4), this solvent combination (CH$_3$CN/HFIP) gave the optimal result. At 60 °C under an ambient atmosphere, polyoxymethylene was completely depolymerized into small molecules within 30 min. Notably, upon the addition of POM (dissolved in HFIP) into the electrolyte

(CH$_3$CN, 0.1 M LiClO$_4$), precipitation of the polymer was observed. However, the resulting suspension quickly turned clear as the electrolysis proceeded (Fig. 2b), and depolymerization products were detected by NMR (see Supplementary Fig. 8).

As indicated by the reaction profile (Fig. 2c), POM was first depolymerized to become formaldehyde and immediately dimerized into oxydimethanol. With the continuation of the electrolysis, formaldehyde trimerizes into its stable form—1,3,5-trioxane. The gradual consumption of formaldehyde/oxydimethanol and the building up of 1,3,5-trioxane eventually led to the formation of formic acid (via anodic oxidation). To achieve higher mass recovery, electro-depolymerization was performed at room temperature (Table 1, Entry 7). Without thermal input, POM underwent complete depolymerization within 2 h, and mass balance was significantly enhanced (~80% mass balance recovered by products I–IV).

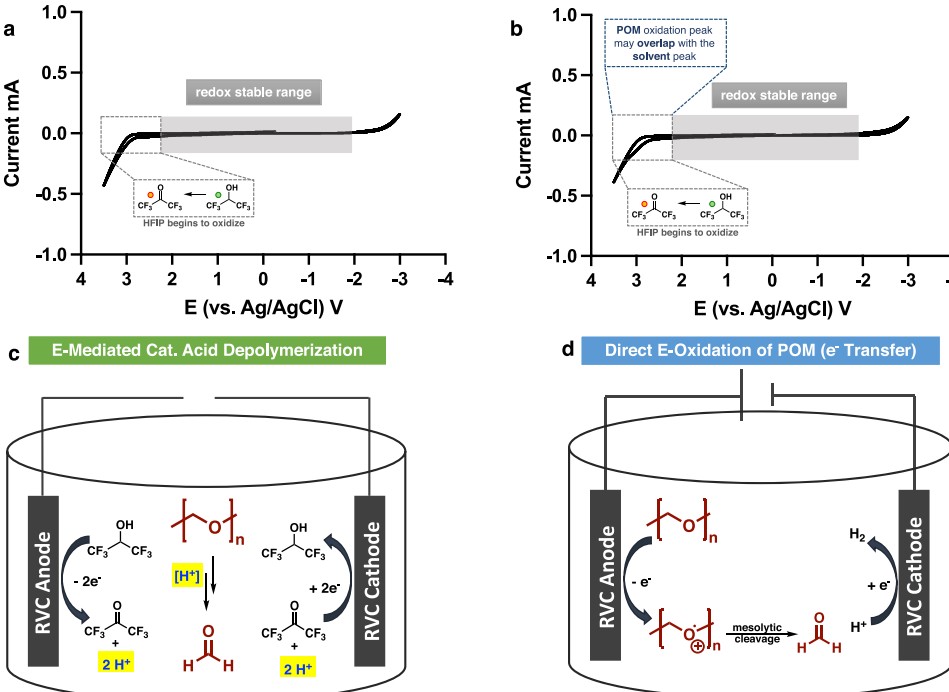

**Fig. 3 | Redox behaviors of HFIP and proposed mechanisms of POM E-depolymerization. a** Cyclic voltammogram of HFIP with 0.1 M TBAClO$_4$ as electrolyte. **b** Cyclic voltammogram of 105 mg of POM dissolved in 10 mL of HFIP with 0.1 M TBAClO$_4$ as electrolyte. (Scan rate 100 mV/s, working electrode: 3 mm glassy carbon, counter electrode: Pt wire, reference electrode: Ag/AgCl). **c** Electro-mediated acid depolymerization of POM. **d** Direct anodic oxidation of POM Note: LiClO$_4$ was found to be insoluble in pure HFIP, CV measurements of HFIP were conducted with TBAClO$_4$.

To ensure the depolymerization process is directly induced by electricity, a control experiment without any applied potential was performed. As shown in Table 1 (Entry 6), without electricity, the polymer does not depolymerize. To ensure the critical role of HFIP, trials without HFIP were conducted at both 60 °C and rt, and depolymerization did not occur (Table 1, Entries 5 and 8).

Therefore, we showed that using appropriate electromediated reaction conditions, POM is efficiently returned into its monomeric form (formaldehyde/1,3,5-trioxane).

## Cyclic voltammetry of HFIP and POM/HFIP

Results from Table 1 illustrated the necessity of HFIP; thus, it is important to investigate the reduction and oxidation processes of both HFIP and POM. Cyclic voltammograms of HFIP were collected first. As shown in Fig. 3a, HFIP showed excellent redox stability. It only began to oxidize with an applied potential greater than 2 V. At the applied potential used in Table 1 (3.5 V), HFIP undergoes fast oxidation to become 1,1,1,3,3,3-hexafluoropropan-2-one—its oxidized ketone form ([19]F NMR see Supplementary Fig. 5). In addition to being a key solvent, HFIP also functions as a catalytic proton donor (2nd role of HFIP) whose proton generation is regulated by the applied potential. More importantly, this result evidently points towards an electro-mediated acid depolymerization mechanism.

Similar voltammogram was obtained (Fig. 3b) for POM solubilized in HFIP (105 mg in 10 mL of HFIP). Like Fig. 3a, oxidation only begins to occur when the applied potential is >2 V. In fact, as a polymeric acetal, the redox potential of POM is expected to be beyond the redox stable window of HFIP. It is unsurprising to observe a similar voltammogram as Fig. 3a. Therefore, this experiment does not eliminate the possibility of a direct electron transfer depolymerization mechanism of POM since in Fig. 3b, the redox peak of POM could be obscured by the broad solvent (HFIP) peaks. Furthermore, this voltammogram revealed a high redox potential ($E_{ox}$ > 2.5 V) for POM oxidation, suggesting it would still require a strong oxidant or oxidizing potential to directly activate the acetal chain.

## Two potential depolymerization mechanisms of POM

On the basis of the results from both solvent screening and cyclic voltammetry, two intrinsically different depolymerization mechanisms were proposed (Fig. 3c, d). The first mechanism is acid-catalyzed depolymerization (Fig. 3c). To test this, we performed a non-electrochemical control experiment (Supplementary Table 1, Entry 4) in which 20 mol% of H$_2$SO$_4$ was combined with the solvent combination (CH$_3$CN:HFIP) used in electrochemical depolymerization. The electro-depolymerization products, except for formic acid, match those in the non-electrochemical depolymerization. However, no depolymerization is observed if the non-electrochemical acid-catalyzed depolymerization is conducted in the absence of HFIP (Supplementary Table 1, Entry 5). As shown in the CV measurement, HFIP functions as an electro-induced proton donor (acid) under anodic oxidation. Taken together, these experiments strongly support an acid depolymerization of polyoxymethylene (Fig. 3c).

The second mechanism considers redox directly on the POM chain. At the applied potential (3.5 V), the possibility of a direct electron transfer mechanism cannot be eliminated since the direct oxidation product—formic acid—was also detected. Anodic oxidation of POM produces the radical cation intermediate; rapid mesolytic cleavage would unzip the polyacetal chain to formaldehyde (Fig. 3d). To identify the preferable E-depolymerization mechanism, divided cell studies, small molecule model studies, experiments with redox-mediators, and low applied potential experiments were undertaken.

## Divided cell studies enabled lower potentials

To identify whether the depolymerization is directly associated with anodic oxidation, a divided cell was used to separate the two electrodes. As shown in Table 2, POM can undergo complete depolymerization in the anode chamber, forming the same products as the

**Table 2 | Divided cell studies**

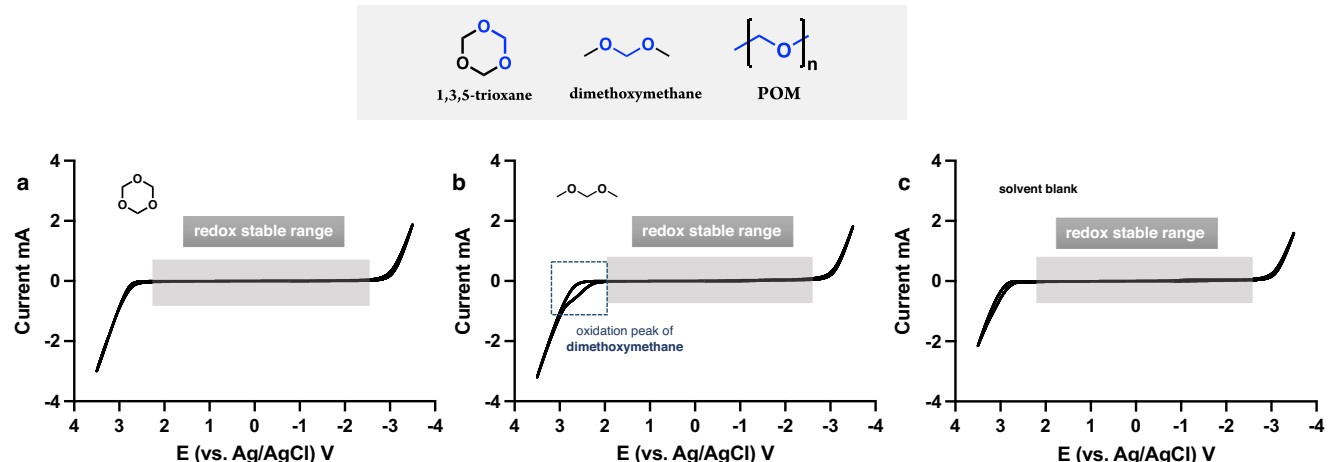

| Entry | Applied potential vs. Ag/AgCl (V) | Working electrode | Solvent | Temperature | Time | Product yield (%) | | | |
|---|---|---|---|---|---|---|---|---|---|
| | | | | | | I | II | III | IV |
| 1 | 3.5 | Cathode | $CH_3CN$:HFIP (4:1) | rt | 1 h | No depolymerization | | | |
| 2 | 3.5 | Anode | $CH_3CN$:HFIP (4:1) | rt | 30 min | 18 | 24 | 34 | 3 |
| 3 | 2.5 | Anode | $CH_3CN$:HFIP (4:1) | rt | 2 h | 13 | 35 | 27 | 1 |
| 4 | 2 | Anode | $CH_3CN$:HFIP (4:1) | rt | 3 h | 17 | 25 | 2 | 0 |
| 5 | 2.5 | Anode | IPA:HFIP (4:1) | rt | 2 h | No depolymerization | | | |
| 6 | 2.5 | Anode | $H_2O$:HFIP (4:1) | rt | 2 h | No depolymerization | | | |

Product yields were determined by $^1$H NMR spectroscopy using mesitylene as standard reference. For details see supplementary information page S7.

**1,3,5-trioxane    dimethoxymethane    POM**

**a** (Cyclic voltammogram of 1,3,5-trioxane)
**b** (Cyclic voltammogram of dimethoxymethane)
**c** (solvent blank)

Current mA vs E (vs. Ag/AgCl) V — redox stable range; oxidation peak of dimethoxymethane

**Fig. 4 | Small molecule model studies. a** Cyclic voltammogram of 1,3,5-trioxane (0.083 M) in 10 mL of $CH_3CN$:HFIP (4:1). **b** Cyclic voltammogram of dimethoxymethane (0.083 M) in 10 mL of $CH_3CN$:HFIP (4:1). **c** Cyclic voltammogram of solvent blank $CH_3CN$:HFIP (4:1) (electrolyte: 0.1 M $LiClO_4$, scan rate: 100 mV/s, working electrode: 3 mm glassy carbon, counter electrode: Pt wire, reference electrode: Ag/AgCl).

undivided cell. This confirms that the electro-depolymerization of POM is directly driven by an anodic reaction. The result strongly supports the two proposed mechanisms in Fig. 3c, d that oxidation was responsible for the depolymerization of the polymer. More importantly, usage of a divided cell was found to enable the further lowering of the applied potential. As shown in Table 2 (Entry 4), POM is fully depolymerized in 3 h at 2 V (vs. Ag/AgCl). Like the results in undivided cell studies, experiments using IPA/HFIP and $H_2O$/HFIP solvent combinations did not trigger the depolymerization of POM (Table 2, Entries 5, 6). Instead, both water oxidation and IPA oxidation were observed. This is surprising; water and IPA oxidation also generate protons, but neither lead to the depolymerization of POM.

Recently, electrolysis with small organic redox mediators is growing in popularity among electrocatalysis, primarily owing to the avoidance of an overpotential[32,39,40]. Thus, experiments with lower applied potentials (-1 V vs. Ag/AgCl) using redox mediators were also performed (for details, see Supplementary Table 4). Both TEMPO and PINO (well-developed organic mediators for electro-oxidation) significantly enhanced the current at low applied potential, but as an oxidizer, they did not trigger the depolymerization of POM. Most likely, it is due to their low reactivities towards the oxidation of HFIP[41].

On the other hand, the successful depolymerization at a lower applied potential in the divided cell also supported the electro-mediated acid depolymerization hypothesis. With an undivided cell, proton formation (HFIP oxidation) and reduction can happen in the

same reaction chamber, and the consumption of the proton at the cathode is likely to compete with the acid-catalyzed depolymerization process. However, in a divided cell, the reduction process is separated from the oxidation. We speculate that this cell configuration allows for the accumulation of the proton in the anodic chamber, thus relaxing requirements for a higher applied potential (faster rate of proton formation).

**1,3,5-Trioxane and dimethoxymethane as small molecule models of POM**

Due to the insolubility of POM in most organic solvents, it is difficult to perform CV measurements directly. Hence, small molecules with identical chemical bonds were used as models to study the redox behavior of the polyacetal. 1,3,5-trioxane, the smallest cyclic form of POM, was chosen as the small molecule representative. Dimethoxymethane, the simplest linear organic acetal, was also used as the small molecule model to study the redox behavior of POM.

As shown in Fig. 4a, cyclic voltammetry indicates that 1,3,5-trioxane is redox stable in the potential window (−3 to 2.5 V). At the same time, this voltammogram is almost identical to that of the solvent redox graph (Fig. 4c), which indicates that 1,3,5-trioxane is less likely to be directly oxidized in this potential range. Similarly, the linear small molecule model, dimethoxymethane (Fig. 4b), is redox stable in the scan range from −3 to 2 V. However, in comparison to 1,3,5-trioxane, dimethoxymethane is redox active when the applied potential is >2 V.

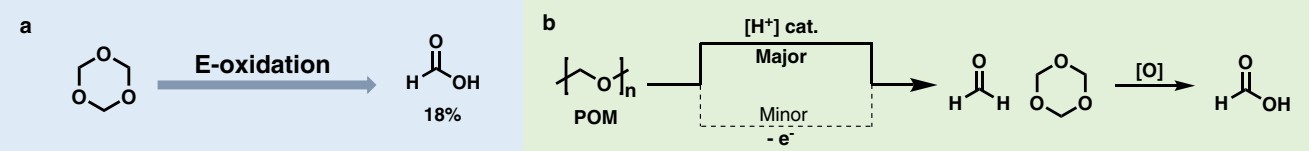

**Fig. 5 | Formic acid formed after POM depolymerization. a** Electro-oxidation of 1,3,5-trioxane to formic acid, reaction conditions: RVC as both working and counter electrodes in a divided cell, with CH₃CN:HFIP (4:1) and 0.1 M LiClO₄ as electrolyte (for details, see supplementary information page S5). **b** Schematic illustration of the major depolymerization pathway of POM.

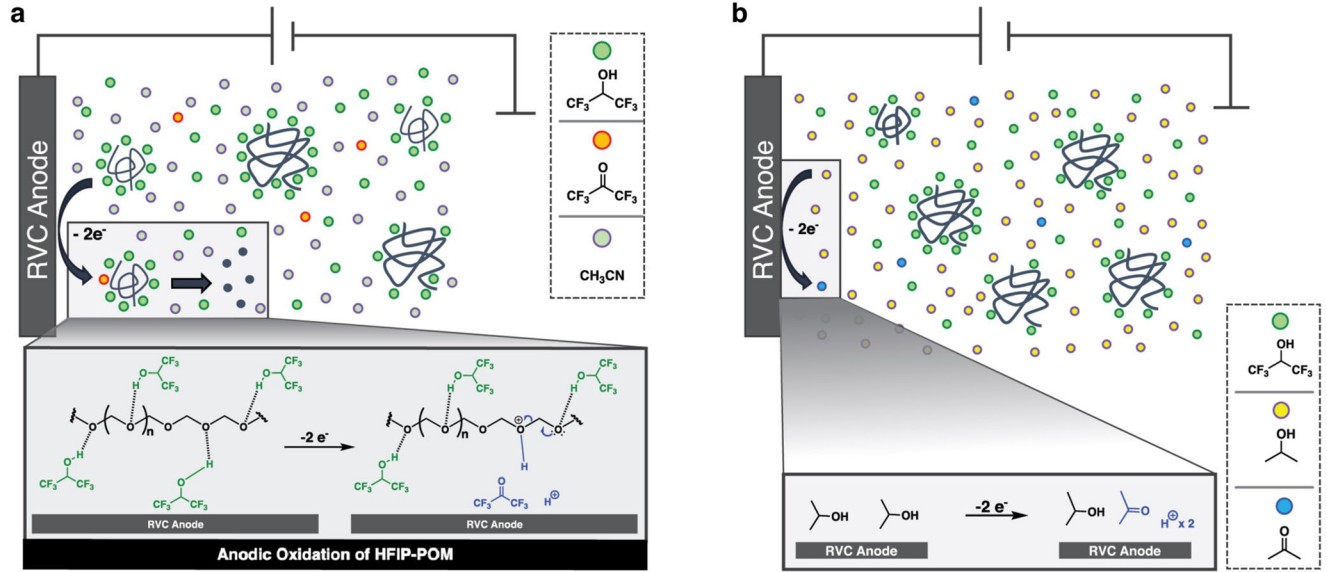

**Fig. 6 | Heterogeneous electromediated acid depolymerization of POM. a** Proton forming from anodic oxidation of HFIP is in closer proximity to the acetal chain. **b** Solvent with a relatively narrow redox-stable window competes with HFIP oxidation.

The voltammogram of dimethoxymethane suggested that it is still possible to have a parallel direct electron transfer mechanism at a potential that is >2 V. However, as shown in Table 2 (Entry 4), POM is fully depolymerized at 2 V, conditions in which direct oxidation is not observed by cyclic voltammetry. Thus, an electromediated acid depolymerization process is more likely to be the dominant breakdown pathway of polyoxymethylene. More importantly, the oxidation product—formic acid—only accumulates after the depolymerization of POM (Fig. 2c). Another key finding obtained from the small molecule studies is the relatively high redox potential of acetals.

### Electro-upgrading of 1,3,5-trioxane: formation of formic acid is independent of the POM depolymerization

Supported by CV measurements, divided cell studies, small molecule model studies, and acid controls, the depolymerization process is likely triggered by the electro-generated proton. However, formic acid was also detected during the electro-depolymerization of POM. To ensure the oxidation product—formic acid—was formed after the depolymerization, electrolysis of 1,3,5-trioxane (major depolymerization product) was performed. As shown in Fig. 5a, under identical electrolysis conditions, 1,3,5-trioxane is directly oxidized to formic acid (for details, see Supplementary Table 7). Combined with the finding in Fig. 2c, formic acid only begins to build up after the complete depolymerization of POM. Thus, the major electro-depolymerization pathway was identified (Fig. 5b). Under applied potential, anodic oxidation of HFIP generates a free proton which catalyzed the depolymerization of the acetal chain. Notice that both products (formaldehyde and 1,3,5-trioxane) do not consume free protons (all C−H were from the original acetal carbon). Therefore, only catalytic

amounts of free protons are needed to completely depolymerize the polymer chain (for details, see supplementary information page S17).

### A heterogeneous electromediated acid depolymerization process: efficient proton transport is the key

As confirmed by both CV and divided cell studies, the overall depolymerization process is mainly triggered by an electro-oxidation generated proton. However, based on the experiments listed in Tables 1 and 2, both water and IPA can also function as electromediated acid donors; yet no depolymerization was observed in either IPA/HFIP or H₂O/HFIP trial. This suggests depolymerization of POM is more nuanced than simply an electro-mediated acid depolymerization. The acid donor is not the only requisite for the depolymerization reaction to proceed. The depolymerization of POM is more likely to be a cooperative process of both anodic oxidation of the solvent HFIP and diffusion of the electro-generated protons.

As elucidated in the beginning, due to the high crystallinity, modification of POM is difficult. A strong H-bonding solvent like HFIP is required to separate the tightly packed polymer chains. Also, due to the extremely low solubility of POM, when the HFIP solvated POM is added to the electrolysis solvent (CH₃CN or IPA), a suspension form (Supplementary Fig. 3). In these heterogeneous electrolyte–polymer mixtures, the diffusion process of the electro-generated proton is critical. Compared to both IPA and water, CH₃CN has a wider redox stable window (similar range as HFIP, see Supplementary Fig. 17a). In this case, HFIP is more likely to be oxidized under the applied potential. Consequently, the HFIP solvated acetal chain is rapidly protonated by such proximity (Fig. 6a), which is hypothesized to assist the chain depolymerization.

**Table 3 | Electro-depolymerization of post-consumer Delrin® waste**

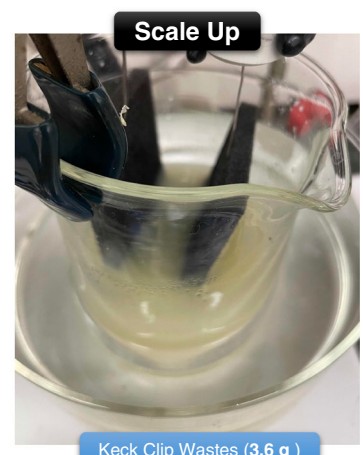

| Entry | Set up | Applied potential vs. Ag/AgCl (V) | Temperature | Time | Product yield (%) | | | |
|---|---|---|---|---|---|---|---|---|
| | | | | | I | II | III | IV |
| 1 | Undivided cell | 3.5 | 60 °C | 1 h | 8 | 27 | 18 | 2 |
| 2 | Divided cell | 2.5 | rt | 2 h | 10 | 14 | 29 | <1 |

Product yields were determined by $^1$H NMR spectroscopy using mesitylene as standard reference (for details, see supplementary information page S7).

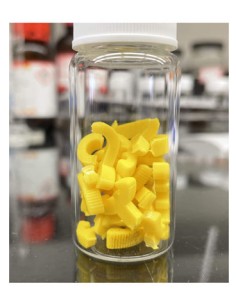

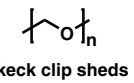

**keck clip sheds**

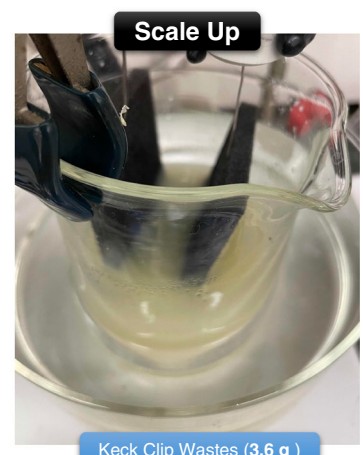

Scale Up

Keck Clip Wastes (3.6 g )

**Fig. 7 | Gram scale depolymerization of commercialized POM.** Pictures of keck clip sheds and the gram scale setup (for details, see supplementary information page S6).

IPA and water have much narrower redox stable windows. Under the same applied potential, oxidation of IPA/water dominated. Despite the constant proton formation, due to the distance between the anode and HFIP-solvated polymer chain, such proton is not effectively transported to the acetal center (Fig. 6b). Compared to HFIP, both IPA and water are more basic, and as such, the generated protons are more likely to be occupied by IPA/$H_2O$ (or their oxidized forms). At the same time, despite its low basicity, HFIP is a strong H-bonding donor. The acetal chain is already partially protonated by HFIP (the solvation process). Therefore, the POM depolymerization is significantly suppressed.

### Electro-chemical depolymerization of post-consumer Delrin® waste and its practical applicability

To test the compatibility of the E-depolymerization method with different additives, Delrin® waste was subjected to our electrochemically mediated depolymerization conditions. As summarized in Table 3, commercialized POM (keck clip shreds) are efficiently depolymerized into small molecules by this electrochemical approach. The observed depolymerization products are found to be the same as the cases using 100% homopolymer (Tables 1 and 2). Based on the successful depolymerization of commercial POM, the practical applicability of this method was also examined. To test the feasibility of this approach in a process setting, we simply conduct the electrolysis in a beaker (open to air) under the same standard reaction conditions (Fig. 7). After 4 h of electrolysis, 3.6 g of keck clip sheds were completely depolymerized, and 34 mol% of monomers were obtained (for details, see Supplementary information page S41). Product recovery is expected to improve by

conducting the electrolysis at ambient temperature, but the depolymerization was significantly slower, which is likely due to the slower mass transport at the concentrations we used in the demonstration (i.e., 5 times higher in polymer concentration compared to the small-scale setup). Overall, this mild electro-depolymerization approach is amenable to the chemical recycling of gram-scale commercialized POM waste.

## Discussion

In summary, a fast and mild electrochemical approach was developed to achieve the post-synthetic recycling of highly crystalline engineering plastic—polyoxymethylene (Delrin®). Upon the solvent screening, hexafluoro-substituted 2-propanol was found to be the potent element during the electro-depolymerization of POM. First, it functions as a potent solvent to break down the orderly packed polymer chains thereby exposing the acetal centers to the following depolymerization process. At the same time, via anodic oxidation, HFIP also functions as a proton donor, which directly triggers the depolymerization of POM. Importantly, effective proton transport in this heterogeneous system appears to be critical. In general, this work successfully demonstrated that electro-mediated redox reactions provide a mild, alternative approach to achieving the post-consumer valorization of an important synthetic plastic. Compared to conventional chemical recycling methods, electrochemical approaches are also more energetically efficient. Post-synthetic electrochemical functionalization of commodity polymers exhibits a promising role in establishing a sustainable chemical future that combines renewable energy and a circular plastic economy. This work may inspire additional research on the electro-deconstruction of other commodity polymers. Future work will target the selective upcycling of POM to formic acid and adopting this method into a flow system.

## Data availability

Details of experimental and analytical procedures and results, control experiments, material characterization, and product determination via $^1$H NMR and $^{19}$F NMR spectra can be found in the supplementary information file or can be obtained upon request from the corresponding author.

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

## Acknowledgements

The research was financially supported by the Joint Center for Energy Storage Research (DE-AC02-06CH11357 to Y.Z., J.S.M. and J.R.-L.), an Energy Innovation Hub funded by the U.S. Department of Energy, Office of Science, Basic Energy Sciences.

## Author contributions

Y.Z. designed, performed, and analyzed the experiments with guidance from J.S.M. and J.R.-L. J.S.M. conceived and directed the project. Y.Z. wrote the manuscript. All the authors participated in the discussion and preparation of the paper.

## Competing interests

The authors declare no competing interests.
