## [Peer Review File · Nature Communications]

Heterogenous Electromediated Depolymerization of Highly Crystalline Polyoxymethylene: A Hydrogen Borrowing ApproachReviewers' Comments:

Reviewer #1:

Remarks to the Author:

In this manuscript, Liu and co-workers developed a novel scheme for electrochemical depolymerization of polyoxymethylene (POM). They found that the reaction proceeded via a heterogeneous electro-mediated acid depolymerization mechanism. The manuscript is well organized and the results are interesting as well. After appropriate modifications, I recommend the publication of this article. Below are my comments:

- 1) Are there any other electrochemical inert electrolytes besides CH₃CN used for the electrolysis of POM?
- 2) Does the type of working electrode have impacts on POM electrolysis?
- 3) CV curves of blank electrolyte without any organic precursors should be illustrated on fig3b and fig4.
- 4) The scan rate of CV seems to be too high.
- 5) Why the applied electrolysis potentials could be decreased when using divided cell?
- 6) To analyze the depolymerization mechanisms of POM, authors carried out a non-electrochemical control experiment. They found out that POM can undergo depolymerization in a CH₃CN:HFIP solution containing 20 mol% H₂SO₄, and the product is consistent with the electrochemical depolymerization process. In this case, is the role of HFIP just to better dissolve POM?
- 7) when using H₂O or IPA as the solvent, although these solvents can also generate protons during the electrolysis, POM does not depolymerize. The author believes that this is due to the difficulty in transporting the electrochemically produced protons to the polymer chains and cannot be effectively utilized. So, if a small amount of acid is added to the aqueous solution or IPA solution to improve the proton transport ability, will the POM depolymerize? If not, then the author's speculated mechanism is wrong. If possible, can it be considered that the purpose of electrolysis is only to generate protons, and can the above solution be used to chemically depolymerize POM? If chemical depolymerization can be achieved then I don't see why electrochemical process is used when simple chemical process can depolymerize POM.
- 8) For the electrochemical system with reference electrode, when introducing the applied voltage, the corresponding reference electrode should be marked, such as 2V vs. Ag/AgCl, instead of 2V.
- 9) In the manuscript, the author should briefly introduce the uses of several products generated by POM degradation.

Reviewer #2:

Remarks to the Author:

The authors introduced their research regarding electromediated depolymerization of POM. The topic is pretty interesting. However, some problems in the manuscript need to be solved before it can be accepted:

1. The authors mentioned the market size and annual growth rate of POM, but a comparison with other polymers is necessary to prove the significance of the depolymerization of POM.
2. The content of Figure 2a is confusing. The only useful information is the aggregation-induced H-bonding enhancement, and one HFIP-POM interaction bond is enough.
3. It is not necessary to demonstrate the co-solvents (IPA, DI-water, DMF) that are not effective for depolymerization in Table 1. The focus should be on CH₃CN. Why choose the CH₃CN:HFIP ratio of 26:4? More experimental results are needed to prove the roles of temperature and time. Also, shouldn't the sum of product yields equal 100%?
4. Figure 2b can be moved to the supplementary section.
5. Figure 2c only illustrates the reaction profile at 3.5V, 60°C, and CH₃CN:HFIP ratio of 26:4. It should be interesting to compare with reaction profiles under other conditions.
6. There is TBAClO₄ in the caption of Figure 3, which was not mentioned in the previous discussions. Please clarify it.

7. In the last paragraph of "Divided Cell Studies Enabled Lower Potentials", the authors mentioned TEMPO and PINO, which were also not mentioned in the previous discussions. Please clarify them.
8. Figure 5a and 5b can be directly expressed as equations in the discussions instead of illustrated in the figures. This can make it more readable.
9. Can this method be applied to other polymers, or at least inspire such applications?

Reviewer #3:

Remarks to the Author:

Zhou et al reports a room temperature electrocatalytic process to convert thermoplastic polyoxymethylene back to a mixture of monomer/trimer. The authors clearly show how the choice of the solvent has a profound effect on the depolymerization and they prove that the acid generated from the anodic oxidation of HFIP is the "active" catalyst. The manuscript flows and reads very well and clearly sets the reader's expectations for the scope of the article. In summary, the work presented here merits publication in Nature Communications and will certainly be of interest to the readership in this journal. However, some points should be addressed:

- 1) Please add the characteristic of the POM used in this study in the main text. What is the Mn, Mw, and PDI?
- 2) Do Mn, Mw, and PDI influence the depolymerization? Is the depolymerization time longer for higher Mn POMs?
- 3) What is the effect of additives and colorant on the depolymerization? The author should comment on this.
- 4) What are the cathode and anode? Why use LiClO₄ as an electrolyte? Please justify and or comment.
- 5) LCA would further improve the quality of the paper. Is the GHG of this process reduced compared to the thermocatalytic counterpart?

REVIEWER COMMENTS

Reviewer #1 (Remarks to the Author):

In this manuscript, Liu and co-workers developed a novel scheme for electrochemical depolymerization of polyoxymethylene (POM). They found that the reaction proceeded via a heterogeneous electro-mediated acid depolymerization mechanism. The manuscript is well organized and the results are interesting as well. After appropriate modifications, I recommend the publication of this article. Below are my comments:

1) Are there any other electrochemical inert electrolytes besides CH₃CN used for the electrolysis of POM?

Response: There could be other solvents useful for POM electrolysis under anodic conditions, but it is important to keep in mind that we require the solvent to be stable also at the counter electrode. CH₃CN is uniquely suited for this purpose. However, to the referee's point, we also performed electro-depolymerization in DMF (another solvent with a stable redox window often used in electrochemical reactions). As shown in Table 1 (Entry 3), no depolymerization was observed. As mentioned, upon electrolysis, DMF oxidation dominated (electrolyte immediately turns yellow and then turns dark brown with extended reaction time).

2) Does the type of working electrode have impacts on POM electrolysis?

Response: Noble metals (such as Pt) were initially considered as an alternative material for the working electrode. Due to its high cost (\$1,110.40 per ounce), we decided to use porous carbon as electrode material.

*Additional experiments using carbon felt and Pt as the working electrodes on the anode side are now in the SI (see page S12). No noticeable differences were observed between different working electrodes since the depolymerization proceeded through a mediated process via solvent oxidation (oxidation of HFIP exhibits similar behavior from different working electrodes [Electrochimica Acta **2012**, 62, 372]).*

3) CV curves of blank electrolyte without any organic precursors should be illustrated on fig3b and fig4.

Response: Addressed. The CV of blank solvent-electrolyte is now found in Figure 4 (Figures 3 and 4 are on adjacent pages).

4) The scan rate of CV seems to be too high.

Response: We are unsure about the point of comparison for the timescale that the referee suggests, but CV experiments at 100 mV/s for electrolyte systems of this type (e.g., comparable viscosity) are standard (see J. Chem. Educ. 2018, 95, 197). Much lower scan rates (e.g., <1 mV/s) are undesirable due to the contribution of natural convection to the electrode response, and at much higher rates (e.g., >10 V/s), the resistance of organic electrolytes increases substantially iR drop. Thus, we decided on a mid-point which seems suitable for our analysis.

5) Why the applied electrolysis potentials could be decreased when using divided cell?

Response: As discussed, we proposed that POM undergoes an electro-mediated acid depolymerization process. The electro-generated protons (via HFIP anodic oxidation) are the key intermediates that catalyze the chain depolymerization. With an undivided cell, proton reduction can happen in the reaction chamber, and the consumption of the proton at the cathode is likely to compete with the acid-catalyzed depolymerization process. However, in a divided cell, the reduction process is separated from the oxidation; we speculate that this allows for the accumulation of the proton in the anodic chamber, thus relaxing the requirements for a higher applied potential (faster rate of proton formation).

6) To analyze the depolymerization mechanisms of POM, authors carried out a non-electrochemical control experiment. They found out that POM can undergo depolymerization in a CH₃CN:HFIP solution containing 20 mol% H₂SO₄, and the product is consistent with the electrochemical depolymerization process. In this case, is the role of HFIP just to better dissolve POM?

Response: The role of HFIP is not only to solubilize the polymer; in the electro-depolymerization process, no acid was added. Protons were generated in situ from the anodic oxidation of the solvent HFIP. As shown in Table 1 (Entry 6), without applied potential (no anodic oxidation of HFIP, also no proton formation), there's no depolymerization even after 24 hours. This suggested the bifunctional role of the vital solvent HFIP.

7) when using H₂O or IPA as the solvent, although these solvents can also generate protons during the electrolysis, POM does not depolymerize. The author believes that this is due to the difficulty in transporting the electrochemically produced protons to the polymer chains and cannot be effectively utilized. So, if a small amount of acid is added to the aqueous solution or IPA solution to improve the proton transport ability, will the POM depolymerize? If not, then the author's speculated mechanism is wrong. If possible, can it be considered that the purpose of electrolysis is only to generate protons, and can the above solution be used to chemically depolymerize POM? If chemical depolymerization can be achieved then I don't see why electrochemical process is used when simple chemical process can depolymerize POM.

Response: The referee raised an important point. This study, on the other hand, does uncover that the electro-mediated depolymerization processes (especially for large macromolecules) are more complicated than those in a homogeneous system, most likely due to the intrinsic heterogeneous nature of electro-mediated chemical processes. In this case, the active intermediate—proton—is formed from anodic oxidation at the anode surface; it needs to diffuse through the polymer chain to access the acetal center. Thus, the surrounding environment (such as solvent) will have a large impact on the proton diffusion process.

Introduction of a small amount of acid (for example, 20 mol% of H₂SO₄) in the presence of water/IPA did not change the outcome. This control experiment is now added in the updated Table S1). However, as we pointed out in the text and depicted in Figure 6, the presence of protons is a necessary but not a sufficient condition for observing POM depolymerization.

This is because, first, the acetal polymer is solubilized by HFIP, owing to its strong H-bonding ability (HFIP displays a bifunctional role). Secondly, considering the presence of large amount of IPA (pKa 17.1) or H₂O (pKa 14), the limited amount of proton would be more likely to be captured by the major solvent (water and IPA) since both were more basic than HFIP (pKa 9.3). Under this circumstance, the limited protons would be occupied by the more basic major solvent (water/IPA), and depolymerization would be suppressed due to the inhibition of proton transport.

To referee's last concern, this electro-mediated depolymerization process does not require direct handling of strong inorganic acid; this is important in industrial processing settings. Since direct usage of strong inorganic acid requires extra safety measurements, and the electrochemical process is more likely to be adapted in a flow system, it also circumvents the construction of large batch reactors.

8) For the electrochemical system with reference electrode, when introducing the applied voltage, the corresponding reference electrode should be marked, such as 2V vs. Ag/AgCl, instead of 2V.

Response: Addressed; see Table column heading.

9) In the manuscript, the author should briefly introduce the uses of several products generated by POM degradation.

Response: Addressed. As discussed in the manuscript, via electro-depolymerization, POM waste can be recycled back to its monomers (formaldehyde and 1,3,5-trioxane). The produced monomer can be subject to a second round of polymerization, which enables the regeneration of materials. Meanwhile, the generated monomers are platform chemicals with established markets for various purposes (synthetic precursor, disinfectant, tissue fixative and embalming agent, etc.)

Reviewer #2 (Remarks to the Author):

The authors introduced their research regarding electromediated depolymerization of POM. The topic is pretty interesting. However, some problems in the manuscript need to be solved before it can be accepted:

1. The authors mentioned the market size and annual growth rate of POM, but a comparison with other polymers is necessary to prove the significance of the depolymerization of POM.

Response: Addressed. Market size of the other four commonly used high-performance engineering plastics is shown in Figure 1b.

2. The content of Figure 2a is confusing. The only useful information is the aggregation-induced H-bonding enhancement, and one HFIP-POM interaction bond is enough.

Response: Addressed. Figure 2a is now updated.

3. It is not necessary to demonstrate the co-solvents (IPA, DI-water, DMF) that are not effective for depolymerization in Table 1. The focus should be on CH₃CN. Why choose the CH₃CN:HFIP ratio of 26:4? More experimental results are needed to prove the roles of temperature and time. Also, shouldn't the sum of product yields equal 100%?

Response: We thank the referee for the suggestion. But the demonstration of other co-solvents (IPA, water, and DMF) is part of the screening process. It's important to compare these results to the case using CH₃CN. Meanwhile, IPA and water also function as proton donors, but they do not have the same result as the more redox-stable CH₃CN. To minimize the amount of HFIP (since it is hexafluorinated), a 26:4 (CH₃CN:HFIP) solvent combination ratio was used. In the divided cell study, a solvent ratio of 4:1 (CH₃CN:HFIP) was used (mainly due to the smaller volume (5 mL) of the divided cell and simpler measure of solvent).

As stated in the manuscript, at 60 °C, the mass balance is significantly smaller than in the room temperature experiments. Further increasing the temperature will result in a significant loss of products. The time effect was shown by the reaction profile (reaction profiles with different conditions were now shown in SI page S18).

Due to the low boiling point of formaldehyde, we are not able to capture 100% of the depolymerized products. But lowering the reaction temperature to room temperature does help to recover most mass loss.

4. Figure 2b can be moved to the supplementary section.

Response: We thank the referee for the suggestion. Figure 2b demonstrates that the reaction is a heterogeneous electro-depolymerization process.

5. Figure 2c only illustrates the reaction profile at 3.5V, 60°C, and CH₃CN:HFIP ratio of 26:4. It should be interesting to compare with reaction profiles under other conditions.

Response: Addressed. Reaction profiles of different reaction conditions are now added in SI Figure S18. As discussed, at lower temperature, depolymerization of POM is slower, but mass balance increased substantially.

6. There is TBAClO₄ in the caption of Figure 3, which was not mentioned in the previous discussions. Please clarify it.

Response: LiClO₄ was found to be insoluble in pure HFIP, and CV measurements of HFIP (Figure 3) were conducted with TBAClO₄ (tetrabutylammonium perchlorate). The caption of Figure 3 is now updated.

7. In the last paragraph of "Divided Cell Studies Enabled Lower Potentials", the authors mentioned TEMPO and PINO, which were also not mentioned in the previous discussions. Please clarify them.

Response: Addressed. Further clarification was now added in the main text. (The redox mediators TEMPO and PINO were used to attempt the electrolysis at lower applied potential; however, they were found unable to activate the oxidation of both polymer and HFIP.)

8. Figure 5a and 5b can be directly expressed as equations in the discussions instead of illustrated in the figures. This can make it more readable.

Response: We thank the referee for the suggestion. Figure 5 is mainly used for a schematic illustration of the major depolymerization pathway of POM. Figure 5a is now updated.

9. Can this method be applied to other polymers, or at least inspire such applications?

Response: Yes, electro-depolymerization as an alternative recycling/upcycling approach has considerable potential to be applied in other commodity polymers (such as polystyrene and other polyacetal). Further discussion is now added to the conclusion/outlook.

Reviewer #3 (Remarks to the Author):

Zhou et al reports a room temperature electrocatalytic process to convert thermoplastic polyoxymethylene back to a mixture of monomer/trimer. The authors clearly show how the choice of the solvent has a profound effect on the depolymerization and they prove that the acid generated from the anodic oxidation of HFIP is the "active" catalyst. The manuscript flows and reads very well and clearly sets the reader's expectations for the scope of the article. In summary, the work presented here merits publication in Nature Communications and will certainly be of interest to the readership in this journal. However, some points should be addressed:

1) Please add the characteristic of the POM used in this study in the main text. What is the M_n , M_w , and PDI?

Response: Due to the severe solubility problem of POM (only found to be soluble in HFIP), a special SEC column (PL HFIPgel phase) is required. The POM homopolymer used in this study was purchased through Sigma-Aldrich; no information on the corresponding molecular weight is provided.

*POM homopolymers (Delrin® and Tenac®) that are currently in the market were produced by two major vendors (DuPont and Asahi Kasei). The weight average molecular weights (M_w) of POM (homopolymer) from both vendors were found to be 149 kg/mol (PDI 2.6) and 137 kg/mol (PDI 2.6) [Int. J. Polym. Sci. **2018**, 7410925 (DOI:10.1155/2018/7410925)].*

The above information is now included in SI (page S2).

2) Do Mn, Mw, and PDI influence the depolymerization? Is the depolymerization time longer for higher Mn POMs?

*Response: The keck clip wastes used in this study are the commercialized copolymer of POM (copolymer: contained 1-1.5 % of -CH₂CH₂O- linkage). Molecular weights of common commercialized copolymers were found to range from 8.6 kg/mol to 100 kg/mol with large dispersity [J. Polym. Res. **2012**, 19, 9775 (DOI 10.1007/s10965-011-9775-3)].*

The average depolymerization time between homopolymer and commercialized copolymer was found to be similar. But the homopolymer, which we believe has a higher molecular weight, does take a longer time to completely solubilize in HFIP at room temperature.

The above information is now included in SI (page S2).

3) What is the effect of additives and colorant on the depolymerization? The author should comment on this.

Response: From the keck clip depolymerization results (see pictures in SI page S39), the yellow color has seemingly remained in the electrolyte after electrolysis. Besides products, no other peaks were found in the NMR spectra. However, for the commercialized POM (keck clip), upon the dissolution of the post-consumer Delrin® plastic, a tiny fraction of solids (we suspect that it is a glass fiber additive) was found insoluble even in HFIP. But in general, we have pointed out there's no major difference in the depolymerization results between the homopolymer and the copolymer with additives.

4) What are the cathode and anode? Why use LiClO₄ as an electrolyte? Please justify and or comment.

Response: Reticulated Vitreous Carbon (porous glassy carbon) is used for both cathode and anode (detailed information was provided in SI). The choice of electrolyte has a minimal impact on the depolymerization results; LiClO₄ was chosen as the electrolyte mostly due to its high solubility in CH₃CN and relatively lower cost compared to TBAClO₄.

5) LCA would further improve the quality of the paper. Is the GHG of this process reduced compared to the thermocatalytic counterpart?

Response: We thank the referee for suggesting a Life-Cycle assessment study and agree on the importance of an LCA. But it is beyond this work's scope; we hope to incorporate the LCA study in our future work (e.g., after adapting into a flow system).

As discussed in the introduction, electro-mediated depolymerization can avoid the direct handling/introduction of strong inorganic acid (e.g., H₂SO₄, mainly produced from fuel combustion in the plant). And compared to traditional thermo-assisted depolymerization, electricity can generate from renewable sources (solar, wind, tide, etc.) We believe this electro-mediated depolymerization approach has a much lower greenhouse gas emission.

Reviewers' Comments:

Reviewer #1:

Remarks to the Author:

The revised manuscript has been well improved, and my questions have been well answered by the authors. I think the revised manuscript can be accepted now.

Reviewer #2:

Remarks to the Author:

The authors have solved some problems or answered some questions I proposed in the previous review. However, there are still some problems that need to be solved before the paper can be accepted:

1. The authors compared the market size of POM and the other four commonly used plastics. It would be more readable if the market sizes were expressed in the same unit (billion USD or million USD).
2. The problems in Table 1 are still not solved: (1) the authors mentioned the use of the solvent ratio of 4:1 had been used in the divided cell, but should also be applied in the undivided cell; (2) different applied potentials, temperatures, and time should be compared respectively while the other conditions keep constant; (3) although the authors mentioned not able to capture 100% of products because of the low boiling point of formaldehyde, the current percentage of less than 80% is not acceptable. The problem of formaldehyde capture needs to be solved.

Reviewer #3:

Remarks to the Author:

All the questions have been addressed point-by-point in the revised manuscript. The current manuscript is appropriate for publication in Nature Communications.

REVIEWER COMMENTS

Reviewer #1 (Remarks to the Author):

The revised manuscript has been well improved, and my questions have been well answered by the authors. I think the revised manuscript can be accepted now.

Reviewer #2 (Remarks to the Author):

The authors have solved some problems or answered some questions that I proposed in the previous review. However, there are still some problems need to be solved before the paper can be accepted:

1. The authors compared the market size of POM and the other four commonly used plastics. It would be more readable if the market sizes are expressed in the same unit (either billion USD or million USD).

Response: Addressed.

2. The problems in Table 1 are still not solved:

(1) the authors mentioned the use of solvent ratio of 4:1 has been used in the divided cell, but should also be applied in the undivided cell;

Response: Addressed. An experiment with a 4:1 solvent ratio conducted in an undivided cell is now included (see Table S6). The depolymerization result is similar to the experiments with a 26:4 solvent ratio.

(2) different applied potentials, temperatures, and time should be compared respectively while the other conditions keep constant;

Response: Addressed. For entries 6 and 8 (control experiments with no depolymerization observed), reaction time is now listed as 2 h, and a note of extended reaction time run is added in the table 1 caption. For entry 4, the reaction conducted at 60 °C, the depolymerization is substantially faster than others (complete depolymerization at 30 min).

(3) although the authors mentioned not able to capture 100% of products because of low boiling point of formaldehyde, the current percentage of less than 80% is not acceptable. The problem of formaldehyde capture need to be solved.

*Response: The overall product yield in the optimized reaction conditions (CH₃CN:HFIP at rt of undivided and divided cells) is around 80%. We believe this is a considerably good mass recovery. Other reported methods of commodity polymer depolymerizations (PS deconstruction under mild reaction conditions) also could not achieve yields greater than 80% (23 mol% [J. Am. Chem. Soc. **2022**, 144, 5745–5749]; 40% [ACS Catal. **2022**, 12, 8155–8163], etc.).*

We understand the importance of maximizing product yields. But often, in process chemistry, especially of potent and robust electrochemical approaches, the final quantified product

*yields from the bulk rarely reach quantitative amount (yields often range from 41-81% [Nat. Chem. **2021**, 13, 367-372]; [Nature **2016**, 533, 77-81]).*

The attempt by using an amine trap to in situ capture the formaldehyde is conducted (page S15). However, due to amine oxidation, depolymerization is inhibited (It also functions as a proton scavenger).

Reviewer #3 (Remarks to the Author):

All the questions have been addressed point-by-point in the revised manuscript. The current manuscript is appropriate for publication in Nature Communications.